# Altered carnitine-acylcarnitine profiles in levothyroxine-treated congenital hypothyroid patients with fatigue: An LC-MS/MS-based study from Bangladesh

Mst. Noorjahan Begum[1,2,3], Suprovath Kumar Sarker[1,2], Md Tarikul Islam[1], Golam Sarower Bhuyan[1], Rumana Mahtarin[1], Mohammad Hridoy Patwary[4], Tasnia Kawsar Konika[5], Syeda Kashfi Qadri[6], Tasnuva Ahmed[7], Hurjahan Banu[8], Nusrat Sultana[1], Asifuzzaman Rahat[1], Kohinoor Jahan Shyamaly[9], Taufiqur Rahman Bhuiyan[7], Mizanul Hasan[5], Mohammad A. Hasanat[8], Abu A. Sajib[2], Abul B.M.M.K. Islam[2], Kaiissar Mannoor[1], Sharif Akhteruzzaman[2], Firdausi Qadri[1,7¤]*

1 Institute for Developing Science and Health Initiatives (ideSHi), ECB Chattar, Mirpur, Dhaka, Bangladesh, 2 Department of Genetic Engineering & Biotechnology, University of Dhaka, Dhaka, Bangladesh, 3 Virology Laboratory, Infectious Diseases Division, International Centre for Diarrhoeal Disease Research, Bangladesh (icddr,b), Mohakhali, Dhaka, Bangladesh, 4 Maternal and Child Health Division, International Centre for Diarrhoeal Disease Research, Dhaka, Bangladesh, 5 Nuclear Medicine and Allied Sciences, Bangladesh Medical University (BMU), Shahbag, Dhaka, Bangladesh, 6 Department of Paediatric Medicine, KK Women's and Children's Hospital, Singapore, Singapore, 7 Mucosal Immunology and Vaccinology, Infectious Diseases Division, International Centre for Diarrhoeal Disease Research, Mohakhali, Dhaka, Bangladesh, 8 Department of Endocrinology, Bangladesh Medical University (BMU), Shahbag, Dhaka, Bangladesh, 9 Department of Pediatrics, Bangladesh Medical University (BMU), Shahbag, Dhaka, Bangladesh

¤ Current address: Infectious Diseases Division, icddr,b, Dhaka, Bangladesh
* fqadri@icddrb.org

## Abstract

Congenital hypothyroidism (CH), characterized by insufficient thyroid hormone production at birth, is frequently associated with fatigue, particularly in cases with delayed diagnosis. This study employed liquid chromatography–tandem mass spectrometry (LC-MS/MS) to profile carnitine and acylcarnitines in late-diagnosed congenital hypothyroid patients receiving levothyroxine (LT4) therapy, with the aim of identifying metabolic alterations that may be associated with fatigue symptoms. A total of 56 late-diagnosed congenital hypothyroid patients and 107 age-, sex-, and BMI-matched healthy controls were enrolled. Blood samples were collected in EDTA tubes and as dried blood spots (DBS) on Whatman® 903 filter paper. LC-MS/MS was used to quantify free carnitine and 28 acylcarnitines, and plasma triglyceride (TG) levels were measured using a biochemical analyzer. Compared to healthy controls, congenital hypothyroid patients showed higher mean (±SD) concentrations of free carnitine (45.38±12.61 vs. 41.54±9.85 µmol/L; P=0.049), total carnitines (67.33±18.27 vs. 62.51±14.13 µmol/L), and total acylcarnitines (21.95±7.66 vs. 20.96±5.61 µmol/L), although only free carnitine levels were statistically significant.

**Data availability statement:** All relevant data are within the manuscript and its Supporting Information files.

**Funding:** This study was partially funded by a grant from the University of Dhaka received from the University Grant Commission (CP-4029). The first author was a Ph.D. student (from 2016 to 2019) under a grant from the University of Dhaka. The Ph.D. program was finished in December 2019. The funders had no role in study design, data collection and analysis, decision to publish, or preparation of the manuscript.

**Competing interests:** The authors have declared that no competing interests exist.

Long-chain acylcarnitines were significantly lower in congenital hypothyroid patients ($2.67 \pm 0.87$ µmol/L) compared to controls ($3.15 \pm 0.93$ µmol/L; $P = 0.0014$). The β-oxidation ratio C0/(C16+C18), a proxy for Carnitine Palmitoyltransferase I (CPT-I) activity, was significantly elevated in patients compared to healthy controls ($34.55 \pm 14.88$ vs. $25.73 \pm 6.87$; $P < 0.0001$). Plasma TG levels were also significantly higher in patients ($88.92 \pm 59.54$ mg/dL) than in controls ($58.33 \pm 15.79$ mg/dL; $P = 0.02$). Metabolic profiling in congenital hypothyroid patients revealed impaired long-chain fatty acid oxidation and elevated triglyceride levels. These metabolic changes may contribute to fatigue symptoms and are potentially associated with reduced CPT-I activity, which is essential for mitochondrial β-oxidation. Additionally, mutations in the *TPO* and *TSHR* genes identified within this cohort may be linked to the observed metabolic alterations. Collectively, these findings suggest a possible interplay between genetic variants, disrupted lipid metabolism, and clinical features of congenital hypothyroidism.

## 1. Background

Hypothyroidism is one of the leading endocrine disorders in which the body cannot produce enough thyroid hormone(s) due to an underactive, missing or ectopic thyroid gland. Congenital hypothyroidism (CH) is defined as the insufficient production of thyroid hormone at birth [1]. Globally, the incidence of CH is approximately 1 in 3,000–4,000 live births, whereas a pilot study in Bangladesh reported a higher incidence of 1 in 1,300 [2,3]. An elevated serum TSH and low $T_4$ or free $T_4$ ($FT_4$) level indicate a hypothyroid condition, which is reversible with appropriate $LT_4$ treatment. Although $LT_4$-mediated management of hypothyroidism has been considered comparatively straightforward, data suggest that 5–10% of $LT_4$-treated patients suffer from persistent clinical complications of hypothyroidism despite the maintenance of normal blood levels of TSH and $FT_4$ [4–6]. Among the physiological distresses in these patients, fatigue and fatigue-related symptoms are the most commonly documented manifestations. However, the mechanism causing persistent fatigue in these patients remains unclear [4,5].

L-carnitine is an endogenous compound and its major biological role is in the transport of fatty acids across the inner mitochondrial membrane for fatty acid oxidation via the reversible binding of acyl groups from Coenzyme A (CoA) [7]. Carnitine also has a role in regulating the cellular-to-mitochondrial ratio of free CoA to acyl-CoA [8]. Alterations in carnitine homeostasis can have a harmful effect on human health and are associated with muscle weakness, progressive cardiomyopathy, and encephalopathy [9–11]. Carnitine supplementation increases fatty acid oxidation and has been used in the management of patients suffering from muscular hypoxia, peripheral vascular disease, angina pectoris, congestive heart failure, and hemodialysis [12–14].

In this study, carnitine and acylcarnitine profiles in whole blood from 56 late-diagnosed congenital hypothyroid patients were analyzed using liquid

chromatography-tandem mass spectrometry (LC-MS/MS). Despite being on a prescribed levothyroxine regimen to maintain serum TSH and $FT_4$ levels within the normal range, these patients continued to experience persistent fatigue-related complications. We hypothesized that these symptoms might be linked to disruptions in carnitine-acylcarnitine homeostasis. The study aimed to comprehensively profile carnitine, acylcarnitine, and triglyceride levels, exploring potential correlations between these metabolic parameters and the chronic fatigue experienced by the patients.

## 2. Methods

### 2.1. Ethics approval and consent to participate

The study received approval from the Medical Research Ethics Committee, National Institute of Nuclear Medicine and Allied Sciences for Human Studies at Bangladesh Medical University (BMU) on 26 April 2017, and the Human Participants Committee, University of Dhaka (CP-4029) on 16 May 2017. Following the ethical approval, we enrolled patients and collected their specimens from 1 June 2017–31 December 2019. Blood samples were collected from the participants with informed written consent from their parents or guardians [15].

### 2.2. Study population

**2.2.1. Patient.** A total of 56 congenital hypothyroid patients with fatigue or fatigue-related symptoms who visited the National Institute of Nuclear Medicine and Allied Sciences (NINMAS) and Department of Endocrinology at Bangladesh Medical University (BMU), Dhaka, for their follow-up examination were enrolled in this present study. Patient recruitment was conducted twice weekly at the Pediatric Endocrinology OPD at BMU and NINMAS, where around 50–60 children present daily with various endocrine disorders. Among them, only 1–2 were confirmed CH cases coming for follow-up. All these patients were late-diagnosed and kept under the treatment of Levothyroxine ($LT_4$). Before enrollment, a written informed consent, along with the clinical information, was obtained from the parent(s) or legal guardian(s) of each patient.

**2.2.2. Healthy controls.** A total of 107 participants (aged 2–18 years) without any thyroid hormone complications and fatigue were enro
lled in this study as the healthy controls. All the participants were residents of Mirpur, Dhaka. A written informed consent was obtained from each healthy participant's parent(s) or legal guardian(s).

### 2.3. Specimen collection and preservation

Blood specimens were collected from both congenital hypothyroid patients and healthy controls four hours post-breakfast. All the patients used to take their morning doses of $LT_4$. Serum TSH and $FT_4$ levels were performed at the hospital laboratory as part of the follow-up examination and the test results were recorded during collection of clinical information. Approximately, 5 mL of blood from each patient and healthy participant were collected in an EDTA vacutainer for plasma separation and a portion of the blood (~75 ul per spot) was deposited on Whatman® 903 generic multipart filter paper (GE Healthcare, Westborough, USA) to prepare a dried blood spot (DBS). The blood spot was dried for at least 4 hours at room temperature. Once dried, the DBS cards were transported to the laboratory and stored at −70°C freezer with desiccant for protection from moisture. All the plasma specimens were stored in a −20°C freezer.

### 2.4. HPLC and MS/MS settings

Free carnitine and a total of 28 acylcarnitines were measured in this study. Quantification of acylcarnitines was done by a Shimadzu LCMS-8050 mass spectrometer (Shimadzu Corporation, Kyoto, Japan) equipped with an electrospray ionization interface in the positive ion mode and multiple reaction monitoring (MRM) system. The overall settings for data acquisition were as follows: The interface voltage at 4.5 kV, interface temperature at 250°C, block temperature at 250°C, heat

block temperature at 400°C, nebulizing gas flow at 3.0 L/min, drying gas flow at 15.0 L/min, collision gas (argon) pressure at 230 kPa and a well time of 5 msec.

## 2.5. Method validation

The LC-MS/MS method used in this study was validated on the Shimadzu LCMS-8050 system (Shimadzu Corporation, Kyoto, Japan) using the NeoMass AAAC kit (Labsystems Diagnostics Oy, Vantaa, Finland). Method validation was performed in accordance with Clinical and Laboratory Standards Institute (CLSI) guidelines (EP5-A2, EP06, and EP17) to ensure the reliability and accuracy of quantification of amino acids and acylcarnitines from dried blood spot (DBS) specimens [16].

Validation was carried out using three levels of quality control (QC) DBS materials—low, medium, and high—provided with the NeoMass AAAC kit. The following parameters were evaluated:

Linearity: All analytes demonstrated strong linearity across the measurement range with coefficient of determination ($R^2$) values >0.99.

Limit of Detection (LOD) and Limit of Quantitation (LOQ): LOD and LOQ were calculated for each analyte according to CLSI EP17 guidelines.

Precision: Intra-assay Coefficients of Variation (%CV): The intra-assay CVs for most amino acids and acylcarnitines were within 20%, except for C6 (36.47%) due to its low abundance in the QC sample (0.13 µmol/L).

Inter-assay Coefficients of Variation (%CV): Ranged between 1.32% and 11.60% for amino acids, and 1.16% to 14.14% for acylcarnitines, well within acceptable limits.

Accuracy: Assessed as relative error (RE%), which ranged from −19.85% to +9.33% for intra-assay and −19.31% to +3.55% for inter-assay measurements.

Recovery: Recovery rates for amino acids ranged from 80.68% to 103.54%, and for acylcarnitines from 93.37% to 108.35%, confirming acceptable performance.

To reduce potential batch effects, samples from healthy controls and clinically suspected patients were randomized and analyzed across multiple batches. Each LC-MS/MS run included low, medium, and high QC controls, ensuring assay performance was continuously monitored and remained consistent throughout the study period.

## 2.6. Sample preparation

Extraction and quantification of free carnitine and acylcarnitines from the DBS cards were done using a NeoMass AAAC kit (Labsystems Diagnostics Oy, Vantaa, Finland) according to the manufacturer's instructions. Briefly, the lyophilized internal standards were reconstituted according to the manufacturer's specifications, and the daily working solution was prepared by diluting the reconstituted internal standards 1:100 (v/v) using the extraction solution. A blood spot with a diameter of 3.2 mm was punched into a U-bottomed microplate well and 100 µL extraction solution with internal standard was added. The 96-well microtiter plate was then covered with adhesive film and incubated for 20 min at room temperature with a shaking speed of 650 rpm. After the incubation time was over, 70 µL of the content from each well was transferred to a V-bottomed analysis plate. The plate was covered with aluminium foil to minimize evaporation of the solution. Finally, the analysis plate was then placed on the auto-sampler of the instrument and 1 µL sample was injected to perform the analysis.

## 2.7. Data collection and data processing

The settings for data acquisition and data processing have been described in detail previously [16]. The concentration of each analyte was measured using the formula: Concentration of an analyte (nmol/mL) = the area of the analyte × the concentration of the internal standard/ the area of the internal standard.

## 2.8. Measurement of plasma Triglycerides (TG)

The plasma triglyceride (TG) levels of both patients and healthy controls were quantified by enzymatic colourimetric assay using Randox Triglycerides assay kit (Randox Laboratories Ltd., Crumlin, UK). The test was performed following the procedure provided in the manual of the kit. The absorbance of the sample was read on a DR-7000D semi-automatic analyzer (Dirui, Jilin, China).

## 2.9. Statistical analysis

Mean, standard deviation (SD), coefficient of variance (CV), and interquartile range (IQR) were calculated using Microsoft Excel (2016). Unpaired t-tests with Welch's correction for patients and healthy controls were performed using GraphPad Prism 5.0 and 10.0 software. To assess the robustness of findings, statistical power was estimated using the pwr package in R (version 4.4.1). P-values were also adjusted for multiple comparisons using the false discovery rate (FDR) method. A P-value <0.05 was considered statistically significant.

## 3. Results

### 3.1. Baseline information of the study participants

The baseline information of all study participants, including patients and healthy controls, were summarized in Table 1. The enrolled congenital hypothyroid patients (N = 56, 27 males; 29 females) had an average age of 7.97 ± 4.29 years and a body mass index (BMI) of 17.0 ± 4.4 kg/m². The healthy subjects (N = 107, 54 males & 53 females) had an age range of 7.24 ± 4.27 years and a BMI of 15.78 ± 2.62 kg/m². As shown in Table 1, the enrollment was age- and sex-matched because there were no significant differences in age and gender distribution between the patients and the healthy control groups. The mean BMI, too, did not differ between the two groups.

All the congenital hypothyroid patients received thyroid hormone replacement therapy on a daily basis, and $LT_4$ dosages were adjusted based on age, sex and BMI of the patients. Dose adjustment was also needed for the congenital hypothyroid patients who had clinical manifestations atypical of hypothyroidism. The mean serum TSH level of the patients before initiation of hormone replacement therapy was recorded 66.07 ± 57.38 mIU/L, whereas the baseline TSH level following $LT_4$ therapy was 3.61 ± 4.41(mIU/L), indicating that there was as much as 18.3-fold higher level of serum TSH before treatment initiation. All the patients enrolled in the study could maintain a normal serum-free T4 ($FT_4$) level (17.68 ± 5.09 pmol/L) following $LT_4$ therapy. However, the study could not compare the $FT_4$ levels of the pre-treatment period with that of the post-treatment period because the former was not recorded for the majority of the patients. All the patients received only $LT_4$ treatment, and none of them had a history of therapy with triiodothyronine ($T_3$) or combination therapy with $LT_4$ and T3.

Table 1. Baseline information of the congenital hypothyroid patients and healthy participants.

| Parameters | Patients | Healthy Controls |
|---|---|---|
| Male/ Female | 27/29 | 54/53 |
| Age (years) mean±SD | 7.97±4.29 | 7.24±4.27 |
| BMI (kg/m²) mean±SD | 17.0±4.4 | 15.78±2.62 |
| Serum TSH (mIU/L) before treatment initiation, mean±SD | 66.07±57.38 | NA |
| Serum TSH (mIU/L) during treatment, mean±SD | 3.61±4.41 | NA |
| *Serum $FT_4$ (pmol/L) during treatment, mean±SD | 17.68±5.09 | NA |
| Average $LT_4$ treatment initiation time in months, mean±SD | 19.7±4.72 | NA |

* $FT_4$ level before treatment initiation was not recorded for the majority of the patients. NA= Not Applicable.

Among 56 enrolled congenital hypothyroid patients, the majority were suffering from dyshormonogenesis (50%, n = 28), whereas about 27% (n = 15), 9% (n = 5), 3% (n = 2) and 2% (n = 1) patients had etiologies of thyroid gland agenesis, thyroiditis, hypoplasia and ectopic thyroid gland, respectively (Table 2). The etiologies of five congenital hypothyroid patients were unknown because these patients did not show positive test results for thyroid radionuclide uptake, thyroid sonography, or serum thyroglobulin, although they were kept on regular LT$_4$ therapy. However, all the patients could persistently maintain normal TSH and FT$_4$ levels upon hormone replacement therapy, although they did have various complications due to late diagnosis and delayed treatment initiation (19.7 ± 4.72 months) (Table 1).

### 3.2. Free carnitine, total carnitine, and total acylcarnitine profile

Despite daily thyroid hormone replacement therapy, the congenital hypothyroid patients had been suffering from persistent fatigue. As carnitine plays a critical role in mitochondrial fatty acid oxidation, it has been suggested that fatigue-related complications may involve altered carnitine-acylcarnitine homeostasis. Table 3 shows the comparison of L-carnitine (C0), total carnitines and 13 long-chain acylcarnitines, as well as total acylcarnitines, between the congenital hypothyroid patients with fatigue or fatigue-related symptoms and healthy controls. The mean (±SD) free carnitine concentration was 45.38 ± 12.61 µmol/L in the patient group, whereas it was 41.54 ± 9.85 µmol/L in the healthy control group, and the difference between the 2 groups generated a P value of 0.049. On the other hand, the concentrations of total acylcarnitines and total carnitines of the patient group were 21.95 ± 7.66 µmol/L and 67.33 ± 18.27 µmol/L, respectively, which were higher than the control group (total acylcarnitine = 20.96 ± 5.61 µmol/L; total carnitine = 62.51 ± 14.13 µmol/L), although the difference did not generate a significant P value.

   **3.2.1. Long chain (LC)-acylcarnitines profile.** In this metabolomic profiling study, 13 long-chain acylcarnitines were analyzed which included 6 saturated (C14, C16, C18, C14OH, C16OH, and C18OH), 5 monounsaturated (C14:1, C16:1, C18:1, C16:1OH, and C18:1OH), and 2 polyunsaturated (C14:2 and C18:2) acylcarnitines. The blood concentrations of acylcarnitines with saturated long-chain fatty acids were 0.09 ± 0.05 µmol/L, 1.02 ± 0.32 µmol/L, 0.39 ± 0.14 µmol/L, 0.01 ± .007 µmol/L, 0.001 ± 0.002 µmol/L and 0.001 ± 0.003 µmol/L for myristoylcarnitine (C14), palmitoylcarnitine (C16), stearoylcarnitine (C18), hydroxymyristoylcarnitine (C14OH), hydroxypalmitoylcarnitine (C16OH) and 3-hydroxystearoylcarnitine (C18-OH), respectively, in the patient's group. The C16, C18 and C16OH acylcarnitine concentrations in the blood of congenital hypothyroid patients were significantly lower than that of the healthy control participants with C16, C18 and C16OH acylcarnitine concentrations of 1.24 ± 0.43, 0.47 ± 0.17 and 0.01 ± 0.003 µmol/L, respectively and the differences between the two groups produced significant P values of 0.0003, 0.002 and 0.001, respectively (Table 3). Adjusted p-values and corresponding power of the significant parameters were shown in Table S1.

   However, C14, C14OH, and C18OH acylcarnitine levels remained almost unaltered in the patients and healthy control groups. On the other hand, the mean concentrations of all the monounsaturated long-chain acylcarnitines

**Table 2. Distribution of congenital hypothyroid patients across hypothyroidism subtypes.**

| Type of thyroid disorder (N = 56) | No. of Patients |
|---|---|
| Dyshormonogenesis | 28 |
| Agenesis | 15 |
| Ectopic | 1 |
| Hypoplasia | 2 |
| Thyroiditis | 5 |
| Unknown* | 5 |

*The type of hypothyroid disorder was unknown because these patients did not undergo thyroid radionuclide uptake, thyroid sonography or serum thyroglobulin test

were almost similar in the patients and the control participants except oleylcarnitine (C18:1). It is noticeable here that oleylcarnitine (C18:1) decreased significantly in the patients (0.7 ± 0.21 μmol/L) compared to the healthy controls (0.81 ± 0.25 μmol/L), generating a P-value of 0.007. Among the two polyunsaturated acylcarnitines, octadecadienyl carnitine (C18:2) was found to be decreased in the patients compared to the controls without generating a statistically significant difference (P = 0.113), whereas the concentration of tetradecadienoylcarnitine (C14:2) of the patients and the healthy controls did not differ. However, in contrast to the short and medium-chain acylcarnitines (Table S2), the mean concentration of these 13 long-chain acylcarnitines was significantly lower in the patients (2.67 ± 0.0.87 μmol/L) than that in the healthy controls (3.15 ± 0.93 μmol/L), and the difference between the two groups generated a significant P-value of 0.0014.

### 3.3. Indicator ratios of metabolic process

With a significant decrease in blood concentrations of long-chain acylcarnitines (C16, C18, C18:1, C16OH and C18OH) along with a total decline in the concentration of LC-acylcarnitines and an increase in the blood level of free carnitine (C0); C0/(C16 + C18) ratio was found to be significantly higher in the patients (ratio = 34.55 ± 14.88, IQR = 12.9, %CV = 43.07) than that in the healthy controls (ratio = 25.73 ± 6.87, IQR = 8.48, %CV = 26.71 (P < 0.0001) (Fig 1a). The C0/(C16 + C18) ratio serves as an indirect marker of CPT-I activity. Our findings may indicate reduced CPT-I activity in the patient group, which is essential for the β-oxidation of long-chain fatty acids, facilitating their conversion into acylcarnitines for transport from the cytosol into the inner mitochondrial matrix [17,18].

**Table 3. Comparison of free carnitine, total acylcarnitines, total carnitines, and long-chain acylcarnitines between the congenital hypothyroid patients and the healthy controls.**

| Parameters | Patients μmol/L (mean±SD) | CV (%) | IQR | Healthy Controls μmol/L (mean±SD) | CV (%) | IQR | P Value |
|---|---|---|---|---|---|---|---|
| **Free Carnitine (C0)** | 45.38 ± 12.61 | 27.79 | 17.45 | 41.54 ± 9.85 | 23.72 | 4.9 | 0.049* |
| **Total acylcarnitines** | 21.95 ± 7.66 | 36.1 | 11.54 | 20.96 ± 5.61 | 25.48 | 6.02 | 0.4 |
| **Total Carnitines** | 67.33 ± 18.27 | 29.43 | 24.92 | 62.51 ± 14.13 | 29.14 | 15.95 | 0.08 |
| **Long chain acylcarnitines** | | | | | | | |
| Myristoylcarnitine (C14) | 0.09 ± 0.05 | 53.0 | 0.04 | 0.09 ± 0.05 | 59.65 | 0.03 | 0.95 |
| Myristoleylcarnitine (C14:1) | 0.07 ± 0.03 | 54.7 | 0.05 | 0.07 ± 0.03 | 46.29 | 0.04 | 0.76 |
| Tetradecadienoylcarnitine (C14:2) | 0.03 ± 0.02 | 72.18 | 0.03 | 0.03 ± 0.02 | 52.0 | 0.02 | 0.95 |
| Palmitoylcarnitine (C16) | 1.02 ± 0.32 | 31.67 | 0.43 | 1.24 ± 0.43 | 34.38 | 0.47 | 0.0003* |
| Palmitoyleylcarnitine (C16:1) | 0.06 ± 0.02 | 34.02 | 0.02 | 0.06 ± 0.02 | 31.03 | 0.02 | 0.58 |
| Stearoylcarnitine (C18) | 0.39 ± 0.14 | 37.95 | 0.19 | 0.47 ± 0.17 | 35.5 | 0.25 | 0.002* |
| Oleylcarnitine (C18:1) | 0.7 ± 0.21 | 30.32 | 0.32 | 0.81 ± 0.25 | 31.43 | 0.3 | 0.007* |
| Octadecadienyl carnitine (C18:2) | 0.29 ± 0.1 | 36.83 | 0.1 | 0.32 ± 0.1 | 31.13 | 0.11 | 0.113 |
| Hydroxymyristoylcarnitine (C14OH) | 0.01 ± .007 | 66.37 | 0.008 | 0.01 ± .006 | 64.17 | 0.008 | 0.76 |
| Hydroxypalmitoylcarnitine (C16OH) | 0.001 ± 0.002 | 71.58 | 0.004 | 0.01 ± 0.003 | 42.13 | 0.005 | 0.001* |
| Hydroxypalmitoleylcarnitine (C16:1OH) | 0.03 ± 0.01 | 51.71 | 0.01 | 0.03 ± 0.014 | 46.96 | 0.01 | 0.26 |
| 3-Hydroxystearoylcarnitine (C18-OH) | 0.001 ± 0.003 | 83.99 | 0.004 | 0.001 ± 0.003 | 59.98 | 0.004 | 0.013* |
| Hydroxyoleylcarnitine (C18:1OH) | 0.01 ± 0.006 | 54.72 | 0.008 | 0.01 ± 0.004 | 38.2 | 0.006 | 0.91 |
| Total long chain acylcarnitines | 2.67 ± 0.87 | 31.02 | 1.13 | 3.15 ± 0.93 | 29.36 | 1.09 | 0.0014* |

CV = Coefficient of variance ; IQR = Interquartile range ; * P< 0.05 was considered significant

Total acylcarnitines = Total long chain acylcarnitines + Total medium chain acylcarnitines + Total short chain acylcarnitines ; Total Carnitine = Free carnitine + Total acylcarnitines

## 3.4. Triglyceride (TG) level

The data demonstrating a disturbance in the metabolism of long-chain acylcarnitines in the congenital hypothyroid patients with fatigue might be due to an insufficient CPT-I activity prompting us to ask whether the long-chain fatty acids are diverted for use in the biosynthesis of TG (Fig 1b). Accordingly, we quantified plasma TG levels. Fig 1b demonstrates that the plasma TG level was significantly higher in the patients ($88.92 \pm 59.54$ mg/dL) (IQR = 77.85, %CV = 66.96) than that in the healthy controls ($58.33 \pm 15.79$ mg/dL) (IQR = 24.75, %CV = 27.06). The difference in TG concentration between the two groups generated a significant P-value of 0.02. A schematic diagram of outcomes was shown in Fig 2. Interestingly, although the TG level in the hypothyroid blood was higher and none of the patients was overweight (average BMI was $17.0 \pm 4.4$ kg/m$^2$), and the findings indicate that the deposition of TG might not have occurred at a significant level in the adipose tissues.

## 4. Discussion

This is the first LC-MS/MS-based metabolic profiling report from Bangladesh analyzing carnitine-acylcarnitine metabolites in the blood of congenital hypothyroid patients receiving thyroid hormone replacement therapy. The study aimed to investigate whether alterations in blood carnitine-acylcarnitine levels in treated congenital hypothyroid patients were associated with fatigue-related complications.

Although the recommended time for diagnosis of congenital hypothyroidism and therapeutic intervention should be initiated between the first two and three weeks of life, the average treatment initiation time for the patients of this study was $19.7 \pm 4.72$ months [19,20]. Clinical complications other than neurologic deficits, if any, in congenital hypothyroid patients are considered reversible upon hormone replacement therapy [21–24]. Unfortunately, all the patients enrolled in this study reported at least one of the hypothyroidism-associated symptoms, with fatigue and fatigue-related symptoms as the most common complications, although they had normal blood TSH and FT$_4$ levels upon daily LT$_4$ treatment. Simply having normal levels of TSH and FT$_4$ may not be sufficient to guarantee adequate thyroid hormone production or response to levothyroxine (LT$_4$) treatment.

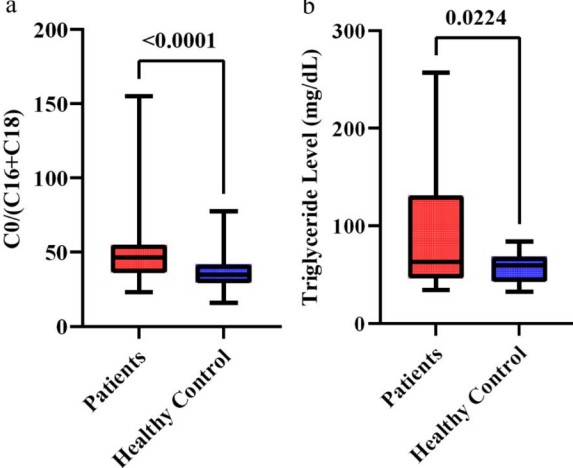

**Fig 1. Comparison of CPT-I enzyme deficiency indicator and plasma TG concentration between the congenital hypothyroid patients and the healthy controls.** (a) C0/(C16 + C18) ratio, an important indicator of CPT-I enzyme activity, was significantly higher in the patients than in the control group (P < 0.0001). (b) Plasma Triglyceride (TG) concentrations were found to be higher in congenital hypothyroid patients compared to the healthy controls, and the difference generated a significant P-value of 0.02. A P-value<0.05 was considered significant.

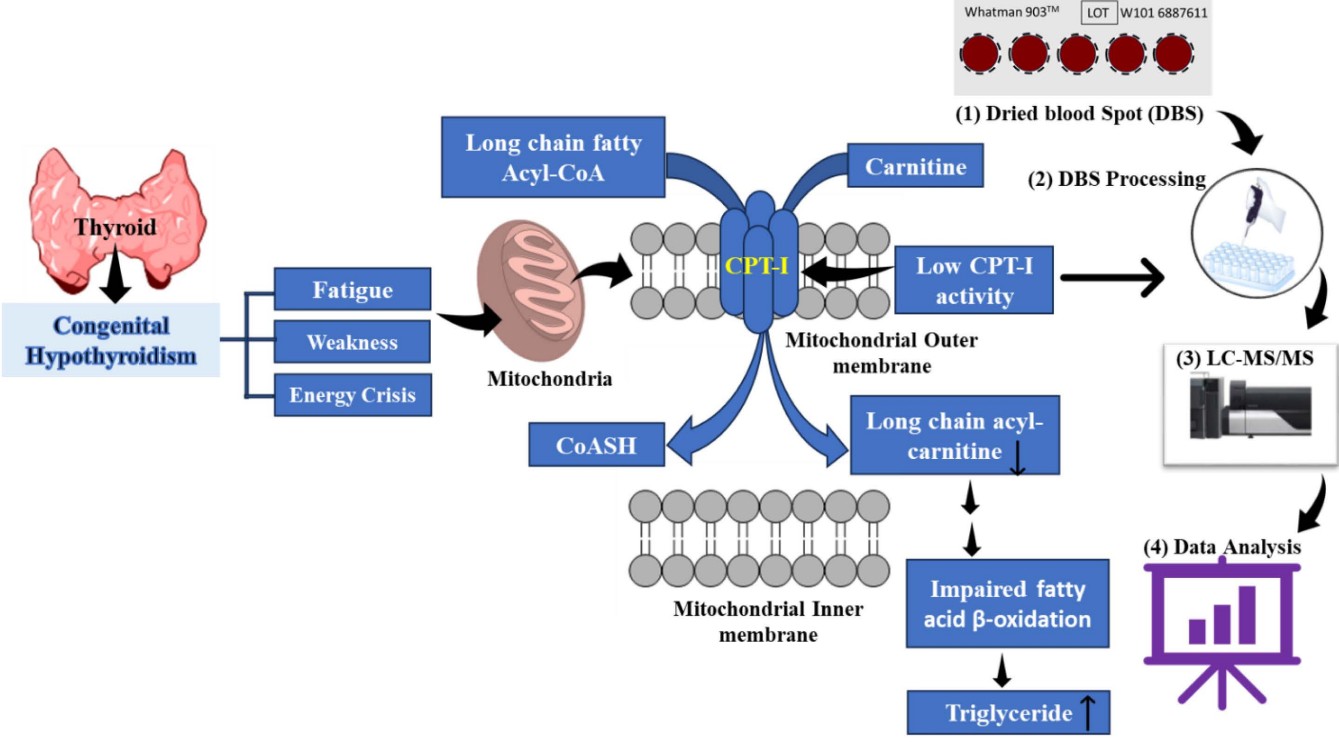

**Fig 2. Schematic diagram of outcomes of metabolic profiling of carnitine, acyl-carnitine, and TG in late-diagnosed congenital hypothyroid patients.**

In this study, the levels of free carnitine, total carnitines, and acylcarnitines were higher in the patient group than in the healthy controls. However, the difference was significant for free carnitine only. Especially the total concentration of long-chain acylcarnitines, along with the concentrations of six individual long-chain acylcarnitines was significantly lower in the patients compared to the controls (Table 3). Wong et al. reported that there were no apparent differences in serum acylcarnitine profiles among hypo-, hyper- and euthyroid states and inferred that the acylcarnitine profiles were relatively unremarkable in thyroid disease [25]. The current study differs from the study by Wong et al. in that we measured carnitine parameters in whole blood using DBS specimens, and the comparisons of the parameters were made between the late-diagnosed thyroid patients under treatment and the healthy controls. In addition, the small sample size and incomplete follow-up were stated as important limitations of their study. However, Galland et al. demonstrated that hepatic carnitine level in rats was modified due to changes in thyroid hormone status, while the carnitine concentrations of kidney and skeletal muscle cells remained unchanged [26]. According to Pande et al., dietary thyroxine could cause an increase in carnitine levels in the liver and serum, while carnitine levels in the heart, skeletal muscle, and the urinary excretion of carnitine remained unaffected in experimental rats [27]. Similarly, thyroxine treatment has also been reported to cause an increase in carnitine in the hearts of guinea pigs [28]. However, a direct comparison of these results with our findings of increased free carnitine in whole blood is difficult due to differences among species as well as variations in the duration and dose of the thyroid hormone replacement therapy. Maebashi et al. found a positive correlation between urinary carnitine excretion and serum thyroxine concentration, with hyperthyroid subjects having significantly elevated levels of carnitine excretion and hypothyroid subjects having markedly reduced carnitine excretion [29]. Urinary carnitine excretion became normal in both groups with the correction of thyroid status. Nevertheless, renal clearance and absorption of carnitine were not

investigated in our study. The ratio of urinary carnitine/acylcarnitine excretion of the patients and healthy controls could have helped us to understand better the altered carnitine-acylcarnitine homeostasis in these late-diagnosed congenital hypothyroid patients under treatment.

Beyond that, this study explored the levels of six Long Chain (LC)-acylcarnitines, as well as the mean concentration of total LC-acylcarnitine was significantly (P = 0.0014) lower in the patients than in the controls, although the levels of short and medium-chain acylcarnitines remained unaffected. A study of chronic fatigue syndrome patients showed a significantly lower level of LC-acylcarnitines compared to the healthy subjects, although there were no significant differences in free carnitine, total carnitine, or total acylcarnitine levels between the two groups [7]. Another study also reported insignificant differences in free carnitine, total carnitine, and total acylcarnitine concentrations between the chronic fatigue syndrome patients and the healthy controls [30]. However, the findings of An et al. made it more paradoxical as they reported that carnitine supplementation could significantly lessen both the fatigue severity scale and mental fatigue score of hypothyroid patients younger than 50 years of age, and thus they concluded that carnitine supplementation may be useful in alleviating fatigue symptoms in this group [31]. Since fatty acid oxidation depends on the presence of carnitine, an increase in the level of carnitine may cause an increase in the rate of fatty acid oxidation and thus contribute to the energy supply. Thyroid hormone controls the hepatic conversion of γ-butyrobetaine to carnitine in rats and also increases carnitine bioavailability [26]. Elevated carnitine levels have been shown to stimulate fatty acid oxidation through mechanisms involving increased CPT-I activity. Interestingly, in this study, although the carnitine level was high in the blood of patients, a decrease in CPT-I activity was found in these congenital hypothyroid patients with energy crises, having normal TSH and $FT_4$ levels after hormone replacement therapy. However, several studies reported carnitine as a peripheral antagonist of thyroid hormone action and may act as a peripheral inhibitor of thyroid hormone uptake [32–34]. This inhibitory effect may be part of a retro-control loop between thyroid hormone and carnitine and thus the reduced thyroid hormone at the cellular level may hinder CPT-I activity [35]. A study of cellular hypothyroidism in different tissues is requisite to minimize the discrepancy between the findings of different study groups and also to devise a proper management strategy for congenital hypothyroid patients with fatigue and related symptoms.

Moreover, a possible decrease in CPT-I activity and reduced β-oxidation—both of which may contribute to an underlying energy deficit in the study patients—prompted us to assess plasma triglyceride (TG) levels to further explore lipid metabolic disturbances.

.In this study, the congenital hypothyroid patients had significantly higher levels of plasma TG, and the finding was supported by other studies [36–38]. The excess TG usually deposits inside the cells of adipose tissues and the liver as lipid droplets [39]. However, none of the patients enrolled in this study was overweight (BMI = 17.0 ± 4.4 kg/m²), and the lean body mass did not differ between the congenital hypothyroid patients and the healthy controls (data not shown), suggesting that the hepatocytes might be the primary site of TG deposition in these patients. A wide range of studies reported both overt and subclinical hypothyroidism as a risk factor of nonalcoholic fatty liver disease (NAFLD) and nonalcoholic steatohepatitis (NASH), the chronic liver diseases that might progress to cirrhosis and hepatocellular carcinoma [37–41 although some newly published studies failed to find such an association [42,43].

It had been demonstrated that acylcarnitine levels were significantly higher in the whole blood than in the plasma because higher concentrations of LC-acylcarnitines in the erythrocyte membranes contributed significantly to the whole blood acylcarnitines [44,45]. However, plasma acylcarnitine concentrations might have a pattern of variation similar to the one observed in the whole blood specimens. Finally, there are a few limitations in the study, notably urinary excretion levels of acylcarnitines of the patients were not measured. Targeting more acylcarnitine metabolites might have helped to delve further into the patients' ongoing fatigue-related complications.

Furthermore, the fatigue commonly observed in these congenital hypothyroid patients may be associated with underlying genetic mutations in the Thyroid Peroxidase (*TPO*) and Thyroid stimulating Hormone Receptor (*TSHR*) genes identified in this cohort. Notably, non-synonymous mutations such as p.Ala373Ser, p.Ser398Thr, and p.Thr725Pro in the

*TPO* gene, along with p.Ser508Leu and p.Glu727Asp in *TSHR* gene, were detected in these patients. These mutations are known to compromise the structural and functional integrity of the respective proteins, impairing their interaction with ligands and regulatory molecules [15,46–48]. Such disruptions in protein function interfere with thyroid hormone biosynthesis and receptor-mediated signaling, resulting in hormonal insufficiency. This deficiency may, in turn, contribute to dysregulated fatty acid metabolism and diminished mitochondrial energy production—metabolic impairments frequently associated with fatigue in hypothyroid states. We hypothesize that reduced CPT-I activity may contribute to persistent fatigue in these congenital hypothyroid patients, even after levothyroxine therapy, which alone may be insufficient to fully address the downstream metabolic consequences of *TPO* and *TSHR* gene mutations present in this cohort. Therefore, addressing inherent defects in hormone synthesis and receptor responsiveness is crucial. Patients with these mutations may benefit from more personalized treatment strategies and close clinical monitoring to ensure optimal thyroid hormone replacement. Moreover, carnitine supplementation could play a supportive role alongside LT4 therapy in mitigating metabolic dysfunction and fatigue.

This study provides valuable insights into metabolic alteration in congenital hypothyroid patients in Bangladesh; however, several factors should be considered when interpreting the findings. The cross-sectional design limited our ability to track treatment outcomes over time, and fatigue was self-reported without standardized severity scoring. While triglyceride levels were assessed, additional metabolic or hepatic markers, including pre-treatment FT4 and urinary carnitine/acylcarnitine levels, were not available. Tissue-specific effects were also beyond the scope of this analysis. Additionally, variability in patient genetics and treatment history may have influenced the observed metabolic profiles.

## 5. Conclusion

This study reveals significant metabolic alterations in congenital hypothyroid patients, including reduced long-chain acylcarnitine levels and inferred low CPT-I activity, which are suggestive of impaired β-oxidation and energy metabolism. These disruptions may contribute to persistent fatigue, even in patients with normalized thyroid hormone levels under levothyroxine therapy. The findings highlight a critical gap in current management strategies that rely solely on hormonal normalization, underscoring the need for more comprehensive care that addresses underlying metabolic dysfunction.

Clinically, these insights suggest that routine metabolic profiling—especially of carnitine-acylcarnitine pathways—could serve as a valuable adjunct in assessing treatment response and guiding individualized therapy. Additionally, these results open avenues for considering adjunct interventions such as carnitine supplementation in selected cases. Future studies, particularly longitudinal and mechanistic investigations, are essential to validate these findings and refine diagnostic and therapeutic approaches for optimizing outcomes in congenital hypothyroidism.

## Supporting information

**S1 Table. Adjusted p-values and corresponding power of the significant parameters from Table 3.**
(DOCX)

**S2 Table. Comparison of 8 short chain and 7 medium chain acylcarnitines between patients and healthy controls.**
(DOCX)

## Acknowledgments

The authors would like to thank the doctors and staff of BMUfor their support in collecting specimens. Special thanks to Md. Yeasir Karim, Infectious Diseases Division, International Centre for Diarrhoeal Disease Research, Bangladesh (icddr,b), Mohakhali, Dhaka-1212, Bangladesh, for his cordial support.

## Author contributions

**Conceptualization:** Mst. Noorjahan Begum, Suprovath Kumar Sarker, Mizanul Hasan, Mohammad A. Hasanat, Abu A. Sajib, Abul B.M.M.K Islam, Kaiissar Mannoor, Sharif Akhteruzzaman, Firdausi Qadri.

**Data curation:** Mst. Noorjahan Begum, Suprovath Kumar Sarker, Mohammad Hridoy Patwary.

**Formal analysis:** Mst. Noorjahan Begum, Suprovath Kumar Sarker, Md Tarikul Islam, Golam Sarower Bhuyan, Mohammad Hridoy Patwary.

**Investigation:** Mst. Noorjahan Begum, Suprovath Kumar Sarker, Md Tarikul Islam, Golam Sarower Bhuyan, Tasnia Kawsar Konika, Syeda Kashfi Qadri, Tasnuva Ahmed, Hurjahan Banu, Nusrat Sultana, Asifuzzaman Rahat, Kohinoor Jahan Shyamaly, Taufiqur Rahman Bhuiyan, Mizanul Hasan, Mohammad A. Hasanat, Abu A. Sajib, Abul B.M.M.K Islam, Kaiissar Mannoor, Sharif Akhteruzzaman.

**Methodology:** Mst. Noorjahan Begum, Suprovath Kumar Sarker, Tasnia Kawsar Konika, Syeda Kashfi Qadri, Tasnuva Ahmed, Hurjahan Banu, Nusrat Sultana, Asifuzzaman Rahat, Kohinoor Jahan Shyamaly, Taufiqur Rahman Bhuiyan, Mohammad A. Hasanat, Kaiissar Mannoor, Sharif Akhteruzzaman.

**Resources:** Tasnia Kawsar Konika, Syeda Kashfi Qadri, Tasnuva Ahmed, Hurjahan Banu, Nusrat Sultana, Asifuzzaman Rahat, Kohinoor Jahan Shyamaly, Taufiqur Rahman Bhuiyan, Mohammad A. Hasanat.

**Supervision:** Kaiissar Mannoor, Sharif Akhteruzzaman, Firdausi Qadri.

**Visualization:** Mst. Noorjahan Begum, Abu A. Sajib, Abul B.M.M.K Islam, Kaiissar Mannoor, Sharif Akhteruzzaman.

**Writing – original draft:** Mst. Noorjahan Begum, Suprovath Kumar Sarker, Md Tarikul Islam, Golam Sarower Bhuyan, Rumana Mahtarin.

**Writing – review & editing:** Mst. Noorjahan Begum, Suprovath Kumar Sarker, Rumana Mahtarin, Mohammad Hridoy Patwary.

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
