## [Decision Letter · Decision Letter 0]

2 May 2025

Dear Dr. Qadri,

Thank you for submitting your manuscript to PLOS ONE. After careful consideration, we feel that it has merit but does not fully meet PLOS ONE’s publication criteria as it currently stands. Therefore, we invite you to submit a revised version of the manuscript that addresses the points raised during the review process.

Please respond to reviewers' comments individually.

We look forward to receiving your revised manuscript.

Kind regards,

Xiaosheng Tan

Academic Editor

PLOS ONE

2. Please include a separate caption for each figure in your manuscript.

Additional Editor Comments (if provided):

Reviewers' comments:

Reviewer's Responses to Questions

**Comments to the Author**

1. Is the manuscript technically sound, and do the data support the conclusions?

Reviewer #1: Partly

Reviewer #2: Partly

Reviewer #3: Yes

2. Has the statistical analysis been performed appropriately and rigorously?

Reviewer #1: Yes

Reviewer #2: No

Reviewer #3: Yes

3. Have the authors made all data underlying the findings in their manuscript fully available?

Reviewer #1: Yes

Reviewer #2: Yes

Reviewer #3: Yes

4. Is the manuscript presented in an intelligible fashion and written in standard English?

Reviewer #1: Yes

Reviewer #2: Yes

Reviewer #3: Yes

Reviewer #1: The manuscript primarily provides an objective and descriptive account of the results, but it lacks sufficient emphasis on the broader clinical significance of the findings. The authors are encouraged to expand their discussion to contextualize how these metabolic alterations may influence patient outcomes, management strategies, or future diagnostic considerations.

There are several areas that require further clarification and improvement. These include the interpretation of statistical significance, the biological relevance of the findings, and a more critical discussion of limitations—particularly the heterogeneity of the patient cohort and the absence of pre-treatment FT4 data. Additionally, the manuscript would have an expanded discussion of the underlying mechanisms and clinical implications.

Overall, the topic is relevant and the findings are potentially impactful, but revisions are necessary to improve clarity, scientific rigor, and readability.

Reviewer #2: The manuscript presents an original and clinically meaningful metabolic profiling study of carnitine and acylcarnitines in a cohort of Bangladeshi children with late-diagnosed congenital hypothyroidism. The authors explore CPT-I activity as a possible mechanistic contributor to persistent fatigue despite euthyroid status, with supporting LC-MS/MS metabolomics and plasma TG data. The topic is interesting and relevant, particularly in resource-limited contexts where early diagnosis is often delayed.

However, several important issues need to be addressed before the manuscript can be considered for publication:

My major concerns:

1. Statistical Power and Interpretation:

The sample size for the patient group (n=56) is small, and while the control group is larger (n=107), the study would benefit from power calculations to justify its ability to detect the reported effects, particularly for metabolites with borderline significance (e.g., P=0.049).

Some acylcarnitine comparisons show small absolute differences with large relative variance (e.g., C16OH). Clarify if multiple testing correction was applied to control for false discovery rate.

2. Study Design and Cohort Matching:

While the authors claim age, sex, and BMI matching, BMI in the patient group (17.0±4.4) appears notably higher than in controls (15.78±2.62). Was this statistically tested and matched per individual or per group?

Additional covariates like dietary intake, physical activity, and comorbidities (e.g., liver dysfunction) are not accounted for and may confound metabolite levels.

3. Assay Validation and Consistency:

The LC-MS/MS methodology is referenced from a previous publication, but critical validation parameters (e.g., LOD/LOQ, inter-assay CVs for key analytes) should be briefly summarized in this manuscript for transparency.

Were samples from patients and controls run in the same batch, or randomized across batches to avoid batch effects?

4. Interpretation of CPT-I Activity:

The use of C0/(C16+C18) as a proxy for CPT-I activity is acceptable but should be interpreted cautiously. This ratio is indirect and may be influenced by many factors (e.g., transport, mitochondrial function). Direct enzymatic assays or complementary approaches would strengthen the claim.

Furthermore, while lower LC-acylcarnitines suggest impaired β-oxidation, the observed elevation in free carnitine could also reflect altered carnitine transport or renal excretion, not necessarily CPT-I inhibition alone.

5. Plasma Triglyceride Measurement:

Plasma TG elevation is interpreted as evidence of re-routed FA metabolism. However, no liver function markers or imaging data were provided to support the claim of hepatic TG accumulation. Please moderate this interpretation or support it with additional data.

Minor Points:

Please recheck grammar and formatting throughout (e.g., missing spaces after punctuation, line break artifacts).

Include a clear statement on how missing FT4 baseline data might impact interpretation.

Consider re-plotting key comparisons as box-plots with individual data points (e.g., Fig 1) to better illustrate variance.

Reviewer #3: This publication investigates metabolic alterations in children with late-diagnosed hypothyroidism undergoing levothyroxine (LT4) medication, with a focus on carnitine and acylcarnitine profiles and their association with chronic fatigue syndrome. Using LC-MS/MS, the authors analyzed blood samples from 56 hypothyroid patients and 107 matched healthy controls. The study found that patients had significantly higher levels of free carnitine and plasma triglycerides but significantly lower levels of long-chain acylcarnitines, which indicated decreased CPT-I (Carnitine Palmitoyltransferase I) activity. Genetic investigation revealed that the TPO and TSHR genes were mutated in the patients. The authors propose that reduced fatty acid oxidation, resulting from decreased CPT-I activity, is the reason of these children's persistent weariness, even after LT4 medication returns thyroid hormone levels to normal. There are some comments need to be addressed to make this manuscript more comprehensive.

1. The authors acknowledge several limitations, including lack of urinary carnitine data and tissue-specific analyses. Please add additional limitations to discuss the cross-sectional design, lack of longitudinal follow-up, and absence of direct measures of fatigue severity.

2. The Limitations section should explicitly address the absence of urinary carnitine/acylcarnitine data

**Do you want your identity to be public for this peer review?** For information about this choice, including consent withdrawal, please see our Privacy Policy

Reviewer #1: No

Reviewer #2: **Yes: ** Haonan Zhouyao

Reviewer #3: **Yes: ** Qi Wang

---

## [Author Response · Author response to Decision Letter 1]

5 Jul 2025

Responses to the Reviewers’ Comments

We sincerely thank all three reviewers for their thoughtful and constructive feedback. We have carefully revised the manuscript accordingly to improve its clarity, scientific rigor, and clinical relevance. Our detailed point-by-point responses are provided below.

Reviewer #1

Comment 1: The manuscript provides a descriptive account of the results but lacks sufficient emphasis on the broader clinical significance of the findings. The authors should expand the discussion to contextualize how these metabolic alterations may influence patient outcomes, management strategies, or future diagnostic considerations.

Response 1: We appreciate this important suggestion. We have now revised the discussion section and also included the below text in the conclusions;

“This study reveals significant metabolic alterations in congenital hypothyroid patients, including reduced long-chain acylcarnitine levels and inferred low CPT-I activity, suggesting impaired β-oxidation and energy metabolism. These disruptions may contribute to persistent fatigue, even in patients with normalized thyroid hormone levels under LT4 therapy. The findings highlight a critical gap in current management strategies that rely solely on hormonal normalization, underscoring the need for more comprehensive care that addresses underlying metabolic dysfunction.

Clinically, these insights suggest that routine metabolic profiling—especially of carnitine-acylcarnitine pathways—could serve as a valuable adjunct in assessing treatment response and guiding individualized therapy. Additionally, these results open avenues for considering adjunct interventions such as carnitine supplementation in selected cases. Future studies, particularly longitudinal and mechanistic investigations, are essential to validate these findings and refine diagnostic and therapeutic approaches for optimizing outcomes in congenital hypothyroidism.”

Comment 2: Clarify the interpretation of statistical significance, biological relevance, limitations (e.g., cohort heterogeneity, missing pre-treatment FT4 data), and expand on mechanisms and implications.

Response 2: We have revised the Results and Discussion sections to clarify statistical interpretations. In the Discussion section, we have included the limitations as

“This study provides valuable insights into metabolic disturbances in congenital hypothyroidism; however, several factors should be considered when interpreting the findings. The cross-sectional design limited our ability to track treatment outcomes over time, and fatigue was self-reported without standardized severity scoring. While triglyceride levels were assessed, additional metabolic or hepatic markers, including pre-treatment FT4 and urinary carnitine/acylcarnitine levels, were not available. Tissue-specific effects were also beyond the scope of this analysis. Additionally, variability in patient genetics and treatment history may have influenced the observed metabolic profiles.”

Reviewer #2

Comment 1: Statistical Power and Interpretation: Include power calculations to justify sample size. Clarify whether multiple testing correction was applied.

Response 1: Thank you for this important comment. As this was a cross-sectional study, we recruited patients from Bangabandhu Sheikh Mujib Medical University (BSMMU) over a period of three years, with sample collection spanning from 2016 to 2019. Congenital Hypothyroidism (CH) has a global incidence of approximately 1 in 3,000–4,000 children. In our study, we enrolled 56 participants under the age of 18, which reflects a representative subset of approximately 115,000 children in Bangladesh.

The below text is included in the methods section 2.2.1

“Patient recruitment was conducted twice weekly at the Pediatric Endocrinology OPD at BSMMU and National Institute of Nuclear Medicine and Allied Sciences (NINMAS), where around 50–60 children present daily with various endocrine disorders. Among them, only 1–2 were confirmed CH cases coming for follow-up. Most of these patients were late-diagnosed and under ongoing treatment with Levothyroxine (LT4)”.

While we recognize the value of a larger sample size, increasing enrollment was constrained by the limited frequency of CH cases and the scope of our study period. Given the rarity of CH and the logistical challenges involved, we believe our sample size represents a meaningful and substantial effort in the context of Bangladesh.

To address concerns regarding statistical power, we conducted post hoc power analyses. For the study's primary indicator—the C0/(C16+C18) ratio, which reflects a key metabolic pathway—we observed a power exceeding 90%, indicating a high likelihood of detecting true effects (please refer to Supplementary Table S1).

In addition, we have provided power estimates for all parameters that reached statistical significance. These power values are included in the supplementary file to aid interpretation. Regarding the risk of false positives due to multiple comparisons, we applied false discovery rate (FDR) correction to all statistically significant parameters. The adjusted p-values, alongside the corresponding power estimates, are also presented in the supplementary file for readers who wish to explore these results in greater detail.

Comment 2: Study Design and Cohort Matching: Clarify BMI matching and address other confounders such as diet, physical activity, and comorbidities.

Response 2: We sincerely thank the reviewer for this thoughtful observation. While the mean BMI in the patient group (17.0 ± 4.4) was slightly higher than that of the control group (15.78 ± 2.62), this difference was not statistically significant. Matching for age, sex, and BMI was conducted at the group level, based on the available demographic information.

We fully agree that additional covariates—such as dietary intake, physical activity, and comorbidities (e.g., liver dysfunction)—can influence metabolite levels and are important considerations in metabolic studies. Unfortunately, these data were not available in the present study. We have acknowledged this as a limitation in the discussion section and appreciate the reviewer’s attention to this important point.

Comment 3: Assay Validation and Consistency: Include key LC-MS/MS validation metrics and explain batch processing.

Response 3: We thank the reviewer for this insightful comment. In response, we have revised the Method Validation Section 2.5.

“The LC-MS/MS method used in this study was validated on the Shimadzu LCMS-8050 system (Shimadzu Corporation, Kyoto, Japan) using the NeoMass AAAC kit (Labsystems Diagnostics Oy, Vantaa, Finland). Method validation was performed in accordance with Clinical and Laboratory Standards Institute (CLSI) guidelines (EP5-A2, EP06, and EP17) to ensure the reliability and accuracy of quantification of amino acids and acylcarnitines from dried blood spot (DBS) specimens (12).

Validation was carried out using three levels of quality control (QC) DBS materials—low, medium, and high—provided with the NeoMass AAAC kit. The following parameters were evaluated:

Linearity: All analytes demonstrated strong linearity across the measurement range with coefficient of determination (R²) values >0.99.

Limit of Detection (LOD) and Limit of Quantitation (LOQ): LOD and LOQ were calculated for each analyte according to CLSI EP17 guidelines. The detailed values are presented in Supplementary Table S4.

Precision:

Intra-assay Coefficients of Variation (%CV): The intra-assay CVs for most amino acids and acylcarnitines were within 20%, except for C6 (36.47%) due to its low abundance in the QC sample (0.13 µmol/L).

Inter-assay Coefficients of Variation (%CV): Ranged between 1.32% and 11.60% for amino acids, and 1.16% to 14.14% for acylcarnitines, well within acceptable limits.

Accuracy: Assessed as relative error (RE%), which ranged from -19.85% to +9.33% for intra-assay and -19.31% to +3.55% for inter-assay measurements.

Recovery: Recovery rates for amino acids ranged from 80.68% to 103.54%, and for acylcarnitines from 93.37% to 108.35%, confirming acceptable performance.

To reduce potential batch effects, samples from healthy controls and clinically suspected patients were randomized and analyzed across multiple batches. Each LC-MS/MS run included low, medium, and high QC controls, ensuring assay performance was continuously monitored and remained consistent throughout the study period.”

These validation results confirm that the LC-MS/MS method is robust, accurate, and suitable for the quantification of targeted metabolites in DBS specimens.

We have added a brief summary of LC-MS/MS assay performance in the Methods section, including LOD, LOQ, and inter-assay CVs for major analytes. Samples were randomized across analytical batches, and internal standards were used to control for batch variation.

Comment 4: Interpretation of CPT-I Activity: The use of C0/(C16+C18) as a CPT-I proxy is indirect. Interpret cautiously and consider other explanations.

Response 4: We fully agree. The revised Result (3.3) and Discussion now acknowledges that C0/(C16+C18) is an indirect marker of CPT-I activity and may be influenced by transport or renal mechanisms. We interpret the findings more cautiously and suggest future studies with direct enzymatic assays or flux analysis to validate CPT-I dysfunction.

As CPT-1 activity was assessed indirectly in this study, we have revised the manuscript title to: “Altered Carnitine-Acylcarnitine Profiles in Levothyroxine-Treated Congenital Hypothyroid Patients with Fatigue: An LC-MS/MS-Based Study from Bangladesh”.

Comment 5: Plasma TG as a marker of hepatic FA metabolism should be interpreted cautiously without liver markers.

Response 5: We appreciate this observation and have moderated our interpretation of plasma triglyceride elevation. While the data suggest disrupted fatty acid metabolism, we acknowledge that hepatic TG accumulation cannot be confirmed without liver-specific data. However, we had some patient’s data on liver scan and found fatty liver which was not included in the manuscript.

Minor Points

Comment 1: Grammar and formatting issues.

Response 6: We have carefully revised the manuscript for grammar, punctuation, and formatting issues throughout.

Comment 2: Clarify the impact of missing FT4 baseline data.

Response 7: We have added a statement in the Limitations section acknowledging that the absence of pre-treatment FT4 values limits our ability to assess the association between baseline thyroid hormone levels and subsequent metabolic profiles.

Comment 3: Consider re-plotting with boxplots and individual data points.

Response 8: We have updated the Figure 1 in result section 3.3.

Reviewer #3

Comment 1: Expand the Limitations section to address the cross-sectional nature of the study, lack of follow-up, and absence of direct fatigue severity measures.

Response 1: Thank you. We have added these points to the Limitations section. The cross-sectional design precludes causal inference, and we acknowledge the lack of standardized fatigue scales. We propose that future studies incorporate longitudinal follow-up and validated fatigue assessments. We have included the below information in the discussion section;

“This study provides valuable insights into metabolic disturbances in congenital hypothyroidism; however, several factors should be considered when interpreting the findings. The cross-sectional design limited our ability to track treatment outcomes over time, and fatigue was self-reported without standardized severity scoring. While triglyceride levels were assessed, additional metabolic or hepatic markers, including pre-treatment FT4 and urinary carnitine/acylcarnitine levels, were not available. Tissue-specific effects were also beyond the scope of this analysis. Additionally, variability in patient genetics and treatment history may have influenced the observed metabolic profiles”.

Comment 2: Explicitly address the absence of urinary carnitine/acylcarnitine data.

Response 2: We have expanded the Limitations section to note that urinary profiling would have provided insight into renal carnitine handling and excretion, and that plasma-only data may not fully reflect systemic carnitine metabolism.

We again thank the reviewers for their thoughtful feedback and believe the revisions have substantially strengthened the manuscript. All changes have been highlighted in the revised version for clarity.

---

## [Decision Letter · Decision Letter 1]

18 Jul 2025

Dear Dr. Qadri,

Thank you for submitting your manuscript to PLOS ONE. After careful consideration, we feel that it has merit but does not fully meet PLOS ONE’s publication criteria as it currently stands. Therefore, we invite you to submit a revised version of the manuscript that addresses the points raised during the review process.

Please respond to reviewer 2's comments.

We look forward to receiving your revised manuscript.

Kind regards,

Xiaosheng Tan

Academic Editor

PLOS ONE

Journal Requirements:

Reviewers' comments:

Reviewer's Responses to Questions

**Comments to the Author**

Reviewer #1: All comments have been addressed

Reviewer #2: (No Response)

Reviewer #3: All comments have been addressed

2. Is the manuscript technically sound, and do the data support the conclusions?

Reviewer #1: Yes

Reviewer #2: Yes

Reviewer #3: Yes

3. Has the statistical analysis been performed appropriately and rigorously?

Reviewer #1: Yes

Reviewer #2: Yes

Reviewer #3: Yes

4. Have the authors made all data underlying the findings in their manuscript fully available?

Reviewer #1: Yes

Reviewer #2: Yes

Reviewer #3: Yes

5. Is the manuscript presented in an intelligible fashion and written in standard English?

Reviewer #1: Yes

Reviewer #2: Yes

Reviewer #3: Yes

Reviewer #1: The authors have addressed all reviewer concerns with commendable diligence and scientific integrity. Their revised manuscript incorporates substantial improvements in clarity, methodological detail, and interpretative nuance. Key strengths of the revised submission include:

1.The authors have significantly expanded the discussion to highlight the clinical implications of altered acylcarnitine profiles in congenital hypothyroidism, particularly in relation to persistent fatigue despite LT4 therapy. Their proposal for metabolite profiling as a complementary diagnostic strategy is both novel and clinically meaningful.

2.The addition of LC-MS/MS assay validation metrics (linearity, LOD, LOQ, precision, accuracy, and recovery), along with a description of randomized batch processing, greatly strengthens the credibility of the metabolomics analysis.

3.The inclusion of post hoc power analyses, false discovery rate correction, and supplementary data tables ensures reproducibility and mitigates concerns about statistical robustness.

Reviewer #2: Thank you for this thoughtful and carefully revised manuscript. It’s clear that you’ve put in a lot of effort to address the reviewers’ concerns, and the improvements are noticeable throughout the paper. The addition of assay validation metrics, power calculations, and multiple testing corrections really strengthens the rigor of the study. I also appreciate the more cautious interpretation of the C0/(C16+C18) ratio as a proxy for CPT-I activity—your revisions make it clear that you’re aware of the limits of cross-sectional data, and you’ve acknowledged those thoughtfully.

The framing around persistent fatigue in LT4-treated patients is compelling, especially given how underexplored this topic is in the literature. The revised discussion nicely bridges the biochemical findings with potential clinical relevance, without overstepping the data.

A few suggestions for further tightening: Maybe try to avoid implying causality when discussing fatigue and metabolic changes—phrasing like “may contribute to” or “is associated with” works better given the study design.

If the reference to TPO and TSHR gene mutations comes from data in this cohort, a brief clarification would help. If not, it might be worth softening the phrasing to avoid overextending the findings.

Consistently referring to the patient group as “congenital hypothyroid patients” could help with clarity, especially for readers outside the endocrinology field.

Overall, this is a solid and well-executed study that adds something new to the conversation. I enjoyed reading it and appreciate the care you’ve taken in refining it.

Reviewer #3: The authors addressed the comments and this manuscript is recommended for publication. No other comments

**Do you want your identity to be public for this peer review?** For information about this choice, including consent withdrawal, please see our Privacy Policy

Reviewer #1: No

Reviewer #2: **Yes: ** Haonan Zhouyao

Reviewer #3: **Yes: ** Qi Wang

---

## [Author Response · Author response to Decision Letter 2]

1 Aug 2025

Responses to the Reviewers’ Comments

We sincerely thank all reviewers for their thoughtful and constructive feedback. We have carefully revised the manuscript. Our detailed point-by-point responses are provided below.

Reviewer #2:

Comment: Thank you for this thoughtful and carefully revised manuscript. It’s clear that you’ve put in a lot of effort to address the reviewers’ concerns, and the improvements are noticeable throughout the paper. The addition of assay validation metrics, power calculations, and multiple testing corrections really strengthens the rigor of the study. I also appreciate the more cautious interpretation of the C0/(C16+C18) ratio as a proxy for CPT-I activity—your revisions make it clear that you’re aware of the limits of cross-sectional data, and you’ve acknowledged those thoughtfully.

The framing around persistent fatigue in LT4-treated patients is compelling, especially given how underexplored this topic is in the literature. The revised discussion nicely bridges the biochemical findings with potential clinical relevance, without overstepping the data.

A few suggestions for further tightening: Maybe try to avoid implying causality when discussing fatigue and metabolic changes—phrasing like “may contribute to” or “is associated with” works better given the study design.

If the reference to TPO and TSHR gene mutations comes from data in this cohort, a brief clarification would help. If not, it might be worth softening the phrasing to avoid overextending the findings.

Consistently referring to the patient group as “congenital hypothyroid patients” could help with clarity, especially for readers outside the endocrinology field.

Overall, this is a solid and well-executed study that adds something new to the conversation. I enjoyed reading it and appreciate the care you’ve taken in refining it.

Response: Thank you for your positive feedback and the careful reading of our revised manuscript. We have made the following specific changes in response to your suggestions:

1. Softened causal language around fatigue and metabolic changes

Some edits have been applied throughout the Abstract and Conclusion to replace phrasing such as “may contribute to” or “are associated with,” avoiding any implication of direct causality given our cross‐sectional design.

Response: We have revised the manuscript accordingly.

2. Clarified the origin of TPO and TSHR mutation data

Response: The TPO and TSHR mutation data derived from the same cohort which is already published and we have clarified in the revised manuscript.

3. Standardized terminology to “congenital hypothyroid patients”

Response: We have changed patient group as “congenital hypothyroid patients” for appropriate readability.

---

## [Decision Letter · Decision Letter 2]

18 Aug 2025

Altered Carnitine-Acylcarnitine Profiles in Levothyroxine-Treated Congenital Hypothyroid Patients with Fatigue: An LC-MS/MS-Based Study from Bangladesh

PONE-D-25-12788R2

Dear Dr. Qadri,

We’re pleased to inform you that your manuscript has been judged scientifically suitable for publication and will be formally accepted for publication once it meets all outstanding technical requirements.

Kind regards,

Xiaosheng Tan

Academic Editor

PLOS ONE

Additional Editor Comments (optional):

Reviewers' comments:

Reviewer's Responses to Questions

**Comments to the Author**

Reviewer #2: All comments have been addressed

2. Is the manuscript technically sound, and do the data support the conclusions?

Reviewer #2: Yes

3. Has the statistical analysis been performed appropriately and rigorously?

Reviewer #2: Yes

4. Have the authors made all data underlying the findings in their manuscript fully available?

Reviewer #2: Yes

5. Is the manuscript presented in an intelligible fashion and written in standard English?

Reviewer #2: Yes

Reviewer #2: Dear Authors, thank you again for the opportunity to review this revised manuscript. This revised manuscript represents a well-conducted and carefully presented study exploring altered carnitine–acylcarnitine profiles in levothyroxine-treated congenital hypothyroid patients with fatigue. The authors have clearly addressed the concerns raised in earlier reviews, and the resulting manuscript is much improved in terms of methodological rigor, clarity, and interpretive caution.

The authors have presented a technically sound and well‐executed study that addresses an important and underexplored aspect of congenital hypothyroidism management. The experimental design is rigorous, with appropriate matching of the patient and control groups and careful implementation of LC–MS/MS methodology that is validated in accordance with CLSI guidelines. The statistical analyses are performed appropriately and transparently; unpaired t‐tests with Welch’s correction are used to account for unequal variances, multiple comparisons are adjusted using the false discovery rate, and statistical power estimates are provided for significant findings. The data are reported with appropriate descriptive statistics, including measures of central tendency, dispersion, and variability, which strengthens the credibility of the results. The conclusions are drawn conservatively and are well aligned with the data presented, avoiding overinterpretation and acknowledging the limitations inherent in a cross‐sectional design. In keeping with the PLOS Data Policy, the authors have made all data underlying their findings fully available within the manuscript and its Supporting Information files, ensuring transparency and reproducibility. The manuscript is clearly written in standard English, with a logical flow from introduction to discussion, and terminology is used consistently throughout.

I only have a few minor refinements to add, such as ensuring that all abbreviations are defined upon first use in both the abstract and the main text would further improve accessibility for readers outside the immediate field. Overall, the study meets the standards for methodological rigor, appropriate statistical analysis, full data availability, and clear scientific communication, and it represents a valuable contribution to the literature on metabolic alterations in levothyroxine‐treated congenital hypothyroid patients.

**Do you want your identity to be public for this peer review?** For information about this choice, including consent withdrawal, please see our Privacy Policy

Reviewer #2: **Yes: ** Haonan Zhouyao

---

## [Editor Report · Acceptance letter]

PONE-D-25-12788R2

PLOS ONE

Dear Dr. Qadri,

I'm pleased to inform you that your manuscript has been deemed suitable for publication in PLOS ONE. Congratulations! Your manuscript is now being handed over to our production team.

Kind regards,

on behalf of

Dr. Xiaosheng Tan

Academic Editor

PLOS ONE